# Optical Micromachines for Biological Studies

**DOI:** 10.3390/mi11020192

**Published:** 2020-02-13

**Authors:** Philippa-Kate Andrew, Martin A. K. Williams, Ebubekir Avci

**Affiliations:** 1Department of Mechanical and Electrical Engineering, Massey University, Palmerston North 4410, New Zealand; k.andrew@massey.ac.nz; 2School of Fundamental Sciences, Massey University, Palmerston North 4410, New Zealand; m.williams@massey.ac.nz; 3MacDiarmid Institute for Advanced Materials and Nanotechnology, Wellington 6140, New Zealand

**Keywords:** optical tweezers, multi-component micromanipulators, radiation damage, life sciences, optical microrobots

## Abstract

Optical tweezers have been used for biological studies since shortly after their inception. However, over the years research has suggested that the intense laser light used to create optical traps may damage the specimens being studied. This review aims to provide a brief overview of optical tweezers and the possible mechanisms for damage, and more importantly examines the role of optical micromachines as tools for biological studies. This review covers the achievements to date in the field of optical micromachines: improvements in the ability to produce micromachines, including multi-body microrobots; and design considerations for both optical microrobots and the optical trapping set-up used for controlling them are all discussed. The review focuses especially on the role of micromachines in biological research, and explores some of the potential that the technology has in this area.

## 1. Introduction

Improvements in tools for the visualisation of objects at the micro- and nano- scale have given researchers the ability to investigate materials and processes previously out of reach. The dominant forces and phenomena that can be accessed at these small scales often differ greatly from those observed at the macroscopic scale. This can lead to challenges, such as adhesion between objects in micro-manipulation experiments [1,2], but also to opportunities, where forces which would be unnoticeable at the macroscale can be used to great effect on micro and -nanoscale objects; such as in the atomic force microscope and optical and magnetic tweezers [3]. Of these technologies, optical tweezers have shown themselves to be particularly flexible tools for the investigation of structures, systems and processes covering the mesoscale. This has led to optical tweezers being used as a tool in the vast fields of biomedical research, material science and soft-matter studies, where they can be used to apply and measure forces that range from 10${}^{-14}$ to 10${}^{-10}$ Newtons. Optical tweezers were invented by Nobel laureate Arthur Ashkin and colleagues and were first officially introduced in a 1986 paper [4] following more than a decade of experiments with what was termed the radiant pressure force [5,6,7,8,9,10]. These experiments were themselves inspired by earlier work by researchers in the early-to-mid 1900s [11,12,13,14], who had experimentally shown what James Clerk Maxwell had theorised: that light carries momentum. However, these earlier researchers had been severely limited by the range of forces that could be applied, and these experiments relied on observation of extremely slight perturbations. This restriction was removed with the invention of the laser [15,16,17], which allowed for experiments with high intensity beams of coherent light. The potential for applications other than simple manipulation of dielectric spheres was soon realised, with a paper by Ashkin and Dziedzic demonstrating the ability to manipulate viruses and bacteria shortly after the influential paper that introduced single-beam optical traps [18]. This led to rapid uptake of the technology by researchers interested in probing the properties of biological subjects. In particular, the relationship between force and biological behaviours has been a subject of enduring research interest, as researchers took note of the ability to investigate intra-cellular phenomena non-invasively [19]. A particularly interesting subset of optical tweezers research is the emerging field of optical microrobotics. Optical microrobotics has been demonstrated to have applications in optical scanning-force microscopy [20], cell manipulation [21,22,23] and cell analysis [24]. The behaviour of optical micromachines has also been used to validate the use of classical mechanical models at the microscale [25,26] as well as providing insight into how material properties may be affected by scale [27]. However, optical micromachines have been under-utilised thus far in biological studies, making this area one with tremendous potential for researchers. In order to place the opportunities for micromachines in context, this review covers a brief introduction to optical tweezers and the impact of optical trapping in biological studies, as well as exploring the concerns associated with optical trapping of biological specimens. The potential for optical micromachines in this area is then explored through an evaluation of the achievements and challenges associated with optical micromachines, considerations for microrobot design and the choice of optical tweezers for manipulation of optical microrobots. Finally, the review finishes with a speculative view of the future, briefly exploring a few possibilities for the use of optical microrobots in biological studies.

## 2. An Introduction to the Theory of Optical Tweezers

Although the theory of optical tweezers is not a central feature of this review, an introduction to the optical force is helpful. Many papers have been written about the theory involved in modelling optical tweezers [28,29,30], and this summary does not hope to offer more than a basic introduction, with the reader encouraged to follow the references for more thorough explanations. While the effect of the sun on direction of comet tails prompted Johannes Kepler’s theories about the mechanical effects of light, the real impact and potential of radiation pressure are most clearly observed on the micro- and -nano-scale, where the intensity afforded by lasers makes the forces originating from changes in the momentum of light significant. The main feature of an optical tweezers setup is a laser focussed through a high numerical aperture (NA) objective, which turns the force associated with the laser into a useful tool. The high NA objective serves to tightly focus the laser beam to a narrow region of space, resulting in an extremely steep gradient of the laser’s electromagnetic field at this point. This is shown in Figure 1. The force associated with the focussed laser beam is treated in two parts in the theory of optial tweezers: a gradient force that “pulls” objects in towards the focus, and a scattering force which acts like water in a “hosepipe” of photons, bombarding the object [31].

Calculations of the gradient and scattering forces can then be performed according to different equations depending on whether the object being trapped can be said to belong to the ray optics regime (particle diameter d >> trapping wavelength λ) or the Rayleigh regime (d << λ). The ray optics explanation assumes that the objects are sufficiently large that they act as lenses, and relies on the understanding that the collimated laser beam is comprised of many single rays of light. The intense focus of the laser beams used for optical trapping means that there is a higher density of rays at the focus. Each ray carries momentum and experiences a change in this momentum when refracted through the microsphere. This change in momentum corresponds to an equal and opposite change in the momentum of the trapped object, acting to move the object. The moment provided by the beam is calculated by decomposing the incident light beam into a series of rays with appropriate intensity, direction and polarisation, and each ray is understood to be a plane wave which changes direction upon interaction with the object’s surface. Diffraction is ignored in this regime, and the gradient (F${}_{grad}$) and scattering (F${}_{scat}$) force can be calculated using Equations (1) and (2), respectively [16,32].
(1)$$F_{grad}=\frac{n_{1}P}{c}Rsin\left(2\theta \right)-\frac{T^{2}\left[sin\left(2\theta -2r\right)+Rsin\left(2\theta \right)\right]}{1+R^{2}+2Rcos\left(2r\right)}$$
(2)$$F_{scat}=\frac{n_{1}P}{c}1+Rcos\left(2\theta \right)-\frac{T^{2}\left[cos\left(2\theta -2r\right)+Rcos\left(2\theta \right)\right]}{1+R^{2}+2Rcos\left(2r\right)}.$$

In these equations the incident momentum is given by *$\frac{\left(n_{1}P\right)}{c}$*where *$n_{1}$*is the refractive index of the trapping medium, *P*is the power of the ray, and *c*is the speed of light. The angle of incidence is given by *θ*and the angle of refraction by *r*. The terms *R*and *T*refer to the Fresnel coefficients for reflection and transmission due to the interface of the medium and target object, which are dependent on the polarisation of the beam [33]. For the case where the light is unpolarised, equal amounts transverse-magnetic (P polarized, $R_{p}$, $T_{p}$) and transverse-electric (S polarized, $R_{s}$, $T_{s}$) are used. The Fresnel coefficients for transverse-magnetic and transverse-electric waves are given by Equations (3)–(6).
(3)$$R_{p}={\left|\frac{\left(n_{1}\sqrt{1-{\left(\frac{n_{1}}{n_{2}}sin\left(\theta_{i}\right)\right)}^{2}}-n_{2}cos\left(\theta_{i}\right)\right)}{\left(n_{1}\sqrt{1-{\left(\frac{n_{1}}{n_{2}}sin\left(\theta_{i}\right)\right)}^{2}}+n_{2}cos\left(\theta_{i}\right)\right)}\right|}^{2}$$
(4)$$R_{s}={\left|\frac{(n_{1}cos\left(\theta_{i}\right)-n_{2}\sqrt{1-{\left(\frac{n_{1}}{n_{2}}sin\left(\theta_{i}\right)\right)}^{2}}}{(n_{1}cos\left(\theta_{i}\right)+n_{2}\sqrt{1-{\left(\frac{n_{1}}{n_{2}}sin\left(\theta_{i}\right)\right)}^{2}}}\right|}^{2}$$
(5)$$T_{p}=1-R_{p}$$
(6)$$T_{s}=1-R_{s}.$$

If the ray optics regime can be thought of as treating objects as lenses, then objects meeting the size requirements to be considered as Rayleigh scatterers can be thought of as particles that become polarised in response to the changing electromagnetic field. The interaction of the induced dipoles with the steep gradient of the electromagnetic field associated with the tightly focussed laser results in the object being pulled towards the waist of the beam [34]. The scattering force can then be understood as being produced because the intensely focussed field is not static, and thus it induces a magnetic field around the dipoles, resulting in the particles being “pushed” [29].

In this regime the equations for the gradient and the scattering forces are given by (7) and (8) [35,36], where *$n_{med}$*is the refractive index of the medium, *c*is the speed of light, *a*is the radius of the object being trapped, *m*is the ratio of the refractive indices of the object and medium (*m $=\frac{n_{obj}}{n_{med}}$*), *I* is the intensity, *r* is the object’s position *(x, y, z)*and *k*is the wave number *(k $=\frac{2\pi }{\lambda }$)*. The intensity of the laser is related to the electromagnetic field *E* by Equation (Equation 9). These equations provide reasonable approximations of the forces acting on spheres ranging from a few nanometres to a few hundred nanometres, but become unreliable as the size of the object of interest approaches the wavelength of the trapping laser [35].
(7)$$F_{\mathrm{grad}}\left(r,t\right)=\frac{2\pi n_{med}a^{3}}{c}\left(\frac{m^{2}-1}{m^{2}+2}\right)\nabla I\left(r,t\right)$$
(8)$$F_{\mathrm{scat}}\left(r,t\right)=\frac{8\pi n_{med}k^{4}a^{6}}{c}\left(\frac{m^{2}-1}{m^{2}+2}\right)I\left(r,t\right)$$
(9)$$I\left(r,t\right)=\frac{\epsilon_{0}cE{\left(r,t\right)}^{2}}{2}.$$

Unfortunately, most of the objects that researchers are interested in trapping have characteristic dimensions that lie between these two basic regimes, in the intermediate range close to the trapping wavelength. Theoretically, for a small, spherical object in a vacuum, Maxwell’s stress tensor provides a solution for the force imparted by an electromagnetic wave, but it assumes complete knowledge of the field, and is impractical for experimental applications. In the case of spherical objects, Mie theory can be used for an exact solution to the problem of scattering [37,38], but it assumes a plane wave, and this makes it inapplicable to the focused beams used for optical tweezers. However, early experiments with laser light validated the use of Mie theory for objects in the intermediate range, with Mie scattering patterns analysed for non-homogeneous, sphere-within-a-sphere particles [39]. This provided valuable information about how this type of particle interacted with the laser light, and provided some evidence that this theory would hold for experiments with biological cells, which cannot always be assumed to be spherical, isotropic or homogeneous structures. Modern use of Lorenz-Mie theory has seen it adapted to cope with arbitrary illumination [40,41], and in this form it is known as Generalised Lorenz-Mie theory (GLMT). GLMT has been combined with other theories of light propagation to more accurately model particular set-ups; such as the Debye-Wolf theory in order to model the forces from holographic optical tweezers [42]. The main limitation of GLMT is that it is applicable to homogeneous, isotropic spheres, rather than complex shapes. Therefore, further generalisation of the theory to extend it to arbitrarily shaped particles was necessary, resulting in the development of T-matrix theory.

First proposed by Waterman in 1965 [43], the T matrix method is useful for calculating the forces on an object in a dynamic optical tweezers set-up, as the matrix depends on the shape, size, refractive index and position of the scattering object, rather than the incident or outgoing fields of light. Additionally, the matrix simplifies for objects with some degree of symmetry, with a sphere being the simplest shape. This greatly reduces the computational effort. The T-matrix method has been reviewed and utilised by many researchers since Waterman’s introduction of the method for solving scattering problems, with different methods for calculating the matrix used [44], including the extended boundary condition method (EBCM) [45] and the point-matching method [46,47]. A comprehensive overview of the T matrix method specifically for optical tweezers has been written by Nieminen et al. [48] and a thorough explanation of T-matrix and generalised Lorenz-Mie theory has also been produced by Gouesbet [49].

While these methods provide a way to calculate the forces associated with optical tweezers, researchers can reduce difficulty by modelling optical tweezers as a Hookean spring [50]. The Hookean spring model is based on the understanding of an optical trap made from a Gaussian beam as being a “harmonic potential well”, with a defined equilibrium point, and a characteristic stiffness constant. Using the separation of the scattering and gradient components of the force as a starting point, in a stable trapping scenario the gradient force will dominate, meaning that the particle will remain at an equilibrium position in a trap, notwithstanding Brownian motion. Then, the stiffness coefficient of the optical trap can be estimated by measuring the root mean square displacement of the particle from the centre of the trap induced by thermal fluctuations and using the equipartition theorem [51,52]. Other calibration methods for measurement of trap stiffness have been developed [53], and are popularly based on balancing a viscous drag imposed on the particle in the trap [54] or by performing power spectrum analysis [55]. The simplicity of the Hookean spring approximation, and the wealth of different methods for calibration hints at a limitation of this method: the effective spring constant is affected by fluctuations in experimental parameters such as viscosity and temperature, and so may require recalibration during long-running experiments. For researchers looking to bypass this issue, a method of force measurement arising from first principles has been successfully demonstrated in both dual-beam, counter-propagating optical traps [56] and single beam traps [57]. This method involves the measurement of the change in momentum of light before and after interaction with the sample, and requires the collection of scattered light in order to extract the change in momentum from the resulting intensity pattern using a position-sensitive detector. A drawback of this method is that it requires precise alignment of all the components required, and it is not feasible to collect all of the scattered light in a single-beam set-up. However, the results from Reference [57] indicate that the "lost" backward-scattered light contributes very little to the overall calculations, and this could be a useful method for researchers working with non-spherical or non-homogeneous objects, as it does not require experiment-specific calibration.

## 3. First Applications and Impact on Fields

While experiments with the radiant pressure force were initially intended for atomic cooling applications [9,31], their potential for use in biological research was quickly realised [18,58,59]. Ashkin’s own experiments with trapping viruses and bacteria [18] sparked further interest in optical tweezers as a non-invasive technique for probing the material properties of biological subjects [60] and for investigating force-related phenomena [61]. In the past few decades this research has included applications that have ranged from investigating the mechanisms behind cell stiffening due to high blood pressure [62] to examining the impact of chemical binding on DNA’s mechanical properties [63,64,65]. In fact, stretching of single-molecules such as DNA has proven to be a topic that has benefited greatly from the development of optical tweezers.

While the pioneering DNA stretching experiment was conducted using magnetic tweezers [66], the spatial resolution of optical tweezers, which, even in the 1990s, allowed displacements of only a few nanometres to be affected [67], proved particularly attractive to researchers for single molecule studies. Additionally, the range of forces that can be applied cover a range from 10${}^{-14}$–10${}^{-10}$ N, which means that several different force regimes can be investigated [68], from those where thermal fluctuations provide restoring forces, to those where high-forces that can impart conformational changes in constituents of the chain [54]. The popularity of DNA stretching studies has meant that there is ample data available for researchers wanting to develop improved models for the molecule’s structure and behaviour [69]. In this way, theoretical models for single-molecule structure and behaviour can be readily evaluated, and their limitations found. An example of this is the breakdown of the freely-jointed model for single DNA molecules and the subsequent validation of the worm-like chain model [69,70,71,72].

Early optical tweezers-based DNA stretches featured molecules that were fixed to a coverslip at one end, with a free-moving end attached to a functionalised microbead. This microbead was then held in the optical trap and the coverslip was translated in order to create a stretch on the molecule [73]. However, this meant that the stretch was affected by disturbances to the translating equipment, due to mechanical coupling of the stage to the rest of the set-up. Additionally, tethering the DNA to the coverslip meant that the stretches could not be truly uni-directional, with an orthogonal component that could not be determined with a level of accuracy comparable to the in-plane displacement. Another popular method for fixing one end of the DNA tether was attaching the corresponding functionalised microbead to a micropipette, which likewise introduces the possibility of unwanted mechanical coupling. A potential fix for the problem of mechanical coupling can be found in the introduction of steerable traps. Steerable traps allow for the manipulation of both ends of a free-floating DNA molecule, decoupling the process from the rest of the apparatus. In the resulting “dumbbell assay”, functionalised beads are attached to DNA molecules that have been modified to feature bead-binding compounds (typically biotin and digoxigenin) at the ends. Each bead is then confined to a separate optical trap, with one trap being stepped away from the other, which is held in a constant position, in order to stretch the molecule [74]. The change in the distance between the beads is taken as the extension of the DNA tether, allowing for the use of the Hookean spring model of optical tweezers. These three different methods for forming a DNA tether for stretching can be seen in Figure 2, which has been reproduced from Heller et al.’s review of the use of optical tweezers for analysing DNA-protein complexes [75].

While the introduction of a second optical trap in DNA stretching studies decoupled the experiment from vibrations of the apparatus, it also introduced another source of intense laser light. It is accepted that laser irradiation has a negative impact on trapped biological samples, but it is not known how much the laser light affects the DNA molecule, as it is not being directly trapped. However, this leads to an important topic; concerns for biological research involving optical tweezers.

## 4. Optical Trapping: Concerns for Biological Research

The negative effects of laser radiation on biological samples were discovered just as quickly as the potential for optical tweezers in biological studies [18], necessitating the development of strategies for minimising damage to biological samples during assays with optical tweezers. The first consideration for making the trapping beam less damaging to samples was the beam wavelength, with the beam changed from visible radiation to infrared in order to reduce absorption [18]. Infrared lasers commonly produce light in the near-infrared range from 700 nm to around 1200 nm. Even within this range there are significant differences in the damage observed in biological samples, depending on the wavelength used [76,77]. Interestingly, while 740–760 nm have been found to be the most damaging, the commonly used 1064 nm wavelength was found to reduce clonability of CHO cells to below 20% of the control group within five minutes of laser exposure [77]. 990 nm, on the other hand, was found by the same study to only reduce clonability to around 70% after a full 20 minutes of exposure, at the same laser power. However, when the damage caused to cellular DNA after exposure for 30s is compared, 1064 nm is clearly less damaging than wavelengths in the range of 700–900 nm [76]. Therefore, the wavelength used for trapping can be clearly seen to be one of the key considerations when trapping biological samples.

In another study, the presence of oxygen was considered as a factor influencing damage, with the motility of E. coli examined to quantify damage [78]. A range of wavelengths from 830 nm to 1064 nm were used, and the wavelength-dependent damage observed appeared to agree closely with that from Liang et al. [77]. When free oxygen was removed from the system, either through introduction of a scavenging species, or by growing and trapping cells in an anaerobic environment, the damage was found to decrease to levels comparable with the control group, which was not exposed to the laser. Other studies have been performed that also seem to corroborate the hypothesis that oxygenation of the sample is a main cause of damage induced during optical trapping [79,80]. However, there is not yet a clear consensus on the exact mechanism for this damage, and other effects related to laser proximity have also been found to induce damage, such as localised heating [81,82]. Research into parameters influencing the damage inflicted by optical trapping has revealed that the trapping medium affects the temperature increase the laser generates, with glycerol producing much higher temperature variation than water [82].

Not all experiments with biological matter can be adapted to remove oxygen, change the trapping medium, or cope with the introduction of a scavenging species. Therefore, other methods for reducing damage as much as possible must be considered. The laser beam itself has been identified as a potential hazard to biological samples, simply due to the high intensity needed to produce an optical trap. As the electromagnetic field varies sharply, with the intensity dropping rapidly with respect to the distance from the waist of the beam, it has been proposed that some of this damage can be avoided simply by spatially separating the optical traps from the object being examined. An example of this theory can be found in the use of functionalised microbeads for DNA stretching, where the DNA itself is not trapped [69] but rather it is chemically bonded to microbeads which act as trapping points for the molecule. The use of these beads is also necessary to enable DNA stretching at all, as the size of the molecule is below the diffraction limit; a limiting factor for optical tweezers until recently [83]. This also has additional advantages, one of which is that the size of the beads can be adjusted to a size that is advantageous for optical trapping with the set-up being used, ensuring a high trap stiffness. The process can also be better visualised with the help of these beads, with the behaviour of the microbeads informing the researcher about the behaviour of the molecule being stretched between them. If direct trapping of the DNA molecule was to be attempted, then the use of a dye would be required, in order to determine the location of the molecule. Use of such dyes can change the mechanical properties of the DNA molecule, depending on the binding mechanism [61], and labelled molecules can be affected by photobleaching, severely limiting experiment times [84]. While using quantum dots as fluorescent markers may solve the issue of photobleaching, there are concerns about toxicity, although this is not considered a drawback for in vitro studies [85].

Larger biological samples such as cells and their organelles can be directly trapped without difficulty [20], but it is desirable to find strategies for mitigating the damage that high intensity laser traps can inflict [86]. As in the case of single-molecule studies, the effect of the wavelength of light used has been considered, with near-infrared traps preferred because of low endogenous absorption at wavelengths between 700 and 1200 nm [87]. Additionally, the use of non-Gaussian traps has been explored as a possible method for stable trapping of living cells. This includes the creation of “optical funnels” using ring-shaped optical fibres to create a funnel of light, in which objects with a lower refractive index than the trapping medium can be stably trapped [88]. Annular traps have also been constructed using axicons, and used to trap living sperm cells, with the added benefit of exposing the spermatozoa to less intense light [89]. Annular traps constructed using axicons have the added benefit of propagating for long distances within the sample, unlike traps constructed using Gaussian beams which are limited to a small distance from the objective. This means that the working distance in the z-direction can be much greater in a ring-shaped trap [90]. The use of two lenses to create the Bessel beams required also means that the size of the resulting ring-shaped beam can be adjusted by moving one of the lenses, changing the focal length and thus the ring’s diameter, which has potential for studies involving objects of different sizes [91]. However, while some experimental success has been demonstrated using axicons for trapping cells and atoms [92], the use of axicon traps is less straightforward than the theory suggests [93], as the lower intensity of the light consequently means that the optical forces produced are lower than in Gaussian-beam traps. This limits the applications of these traps, as they cannot produce the high forces required for applications such as investigating the mechanical properties of cells [94]. This means that alternative solutions that are not reliant on changing the power or intensity of the trapping beam are desirable for applications that require high forces.

A possible method for retaining the effective power of optical traps, while reducing damage to the subject is using other optically trapped objects as intermediates, effectively using them as end-effectors to manipulate delicate biological subjects. The use of functionalised beads for DNA stretching can be regarded as pioneering work in indirect optical trapping, with functionalised rigid-body structures made from SU-8 also used to manipulate cells in a system with six degrees of freedom [21]. Indirect optical manipulation of cells has also been demonstrated using non-functionalised end-effectors [22], indicating that non-specific bonding due to physical adhesion forces can be sufficient for such studies. This indicates that intermediate optically trapped objects may have potential for a variety of applications where orientation of the cell is important, such as cell aspiration and nucleus extraction. Additionally, experiments using trapped objects as “force probes” indicate that optical end-effectors can be designed to retain the sensitivity of force application that optical tweezers are known for [95,96]. This is important, as the range and resolution of force that can be applied by optical tweezers are distinguishing features of the technology when compared to conventional mechanical manipulation, where limited resolution of force and displacement can limit the use of these methods for high-precision tasks [97].

## 5. Optical Microrobots: Potential Tools for Biological Studies

While researchers have not yet reached a consensus on the exact mechanisms for damage incurred during optical trapping, it is widely agreed that exposure to intense light is a problem. As previously mentioned, it is not always possible to adapt experiments for lower trapping powers, and it is not always practical for researchers to have multiple custom optical trapping set-ups for different samples. Therefore, using microrobots to reduce specimen contact with the trap’s focus, while still enabling application of optical trapping forces, would be a conceptually simple solution for the problem of optically inflicted damage. As already noted, the use of optically trapped objects as scanning probes, for DNA stretching, and as tools for simple cell rotation can be thought of as pioneering work in the field of optical microrobotics. It follows that the use of more complex, articulated robots may be the solution to reducing exposure to laser light, and potentially they may also be the answer to overcoming the limitations of optical tweezers, such as limited force and difficulty performing truly 3D manipulation.

While the potential for optical microrobotics in biological research is clear, there are still relatively few examples of the technology used in this field. This could be partially attributed to historical difficulties in manufacturing optical microrobots. Micro-manufacturing methods have traditionally been iterative processes based off those used for semi-conductor applications, limiting the resolution and shapes that can be achieved. While there is some scope for the use of these iterative-process methods in the development of microrobots [98], they are limited due to their two-dimensional nature. In the case of the paper by Mittas et al., the difficulty of using traditional silicon micromanufacturing techniques can be seen due to the need of precise alignment of the different patterning stages. Additionally, the use of etching for high aspect-ratio features is problematic as it is a time dependent process, which means that the etchant generally spreads in an undesirable manner, leading to feature-undercut. However, this undercutting has been purposefully used to produce three-dimensional micromachine elements, when etchant has been used to remove patterned elements from silicon wafers to produce independent components for use with optical tweezers [99]. In situations where repeatable manufacturing and high feature resolution (<1 μm) are required, relying on etching to produce the desired outcome becomes impractical. Similarly, photoresist patterning using mask aligners can be used to good effect to produce reasonably complex structures [100], but this is an arduous task that is prone to error due to the need for extremely precise alignment of subsequent layers, and only allows for 2.5 dimensions, with extrusion of a planar pattern occurring in one direction. Additionally, making multi-component robots from these methods is complex, due to the requirement for assembly, although the components themselves can be produced [101]. Fortunately, the development of three-dimensional photolithography, based on two-photon absorption polymerisation, has enabled the manufacture of complex structures with <100 nm resolution. The use of this laser-based 3D printing technology is well-established, with the first patent for 3D stereolithography granted in 1986 [102], and examples of complex structures present in literature since the early 2000s [103].

Although two-photon absorption was first theoretically suggested by Nobel laureate Maria Göppert-Mayer in her doctoral thesis [104], like many light-based phenomena it was only experimentally demonstrated after the invention of the laser. Two-photon absorption involves the absorption of two photons- that can be of the same or different energies- to bridge the energy gap between electronic states of a molecule and bring it into a higher energy electronic state. The especially remarkable feature of two-photon absorption is the speed at which the second photon must be absorbed to prevent the molecule in question returning to its initial state from the intermediate, higher-energy virtual state it occupies following the absorption of the first photon. The fact that two photons must be absorbed for the process to occur means that it is non-linear, with the rate of absorption (*R*) being given by (10), where *δ*is the absorption cross-section and *I*is the intensity of the light.
(10)$$R=\delta I^{2}.$$

This phenomenon is the cause of two-photon absorption polymerisation (TPAP), which occurs when two-photon absorption occurs in a matrix of monomers or oligomers and a photoacid-generator [105]. This initiates polymerisation of the material in the regions of the material where the required energy threshold is met [106]. A highly focussed, femto-second pulsed laser beam is used to provide the necessary light for the reaction, and due to the narrow region of focus, high aspect ratio features with feature resolution of less than 100 nm is possible, as the region of polymerisation can be finely controlled [107,108]. This has enabled the printing of complex structures with fine detail, with functional 3D-printed helical springs being evidence of what the technology can achieve [24,109]. This technology is especially promising for creating optical microrobots for use in biological studies, as several commercial resists are biocompatible, with SU-8 being a common example. Additionally, conceptually any resin can be used, so long as it has sufficient absorption of the wavelength of the laser used to induce TPAP, and a high enough proportion of photoacid in the matrix. This allows researchers to create custom resins with the properties that they require, a feature that has already been exploited by researchers creating cell-scaffolds from hydrogels [110]. While in principle two-photon absorption polymerisation simply requires a high-speed pulsed laser, the invention of the Nanoscribe, a commercial solution developed by researchers at Karlsruhe Institute of Technology, has led to more user-friendly TPAP-based 3D printing. The availability of commercial solutions such as this means that 3D laser printing has become accessible to researchers from different backgrounds, widening the scope of applications.

## 6. Optical Microrobots: Achievements to Date and Challenges

It has been established that complex shapes can be printed using TPAP, and there has been some work performed using optically trapped objects as end-effectors for studies involving the manipulation of biological subjects [111]. This positions optically driven micromachines as a promising technology for researchers looking to reduce exposure to intense laser traps. However, there are also several challenges that need to be addressed before the technology can be adopted for regular use, rather than as a novelty.

Optically-driven microrobots have been successfully used to amplify forces from optical tweezers [24,25], and to transform planar input-motion to out-of-plane rotation [112]. It has also been demonstrated that trapped microspheres, arranged in groups of three, create an attractive pocket that approximates a weak optical funnel; which could be used to indirectly trap biological objects [113]. This is similar to what can be achieved using the “doughnut” shaped beams used for dark trapping [88], and demonstrates the potential for optical microrobots as an alternative and more flexible solution to altering the beam used for optical trapping. Additionally, the impressive resolution and dimensions that can be achieved using TPAP has enabled the printing of nano-springs. These nano-springs have been used as more than just examples of what TPAP can achieve, they have also been used to measure force amplification [24], and to investigate the impacts of post-curing UV exposure on photoresins [114], using optical tweezers. The use of levers and articulated microrobots to amplify force and produce out-of-plane movement shows that micromachines can be used to improve upon the abilities of optical tweezers, which are currently rather limited in these areas. Additionally, by demonstrating that classical models of springs hold at the microscale [24,26], researchers have shown that springs could potentially be used to directly measure the forces applied with optical tweezers. However, this approach requires extensive knowledge of the polymer used to fabricate the springs and precise control of printing parameters, as non-uniformity could lead to large differences from expected results [115]. This has become less of a problem as laser-based 3D printing technology has advanced, and as more research has been performed to ascertain the effects that laser printing parameters have on mechanical properties of printed objects [116], but it is still worth considering.

While these examples show what can be achieved in optical microrobotics, the authors also make note of the challenges involved in producing and actuating them. Particular difficulties are caused by adhesion forces between discrete moving parts [25,112] and the tolerance of pin joints when optical trapping is performed. Adhesion forces in particular form a barrier when it comes to developing articulated optical microrobots, as more complex machines offer more opportunities for seizure due to links and joints bonding to each other through physical interactions such as van der Waals forces. Additionally, this poses a particular barrier for the use of microrobots in biological research, where studies are often carried out in particular fluids which are intended to mimic in vivo conditions. Some of these fluids, such as the tris-buffered saline used for DNA stretching [117], have relatively high salt concentrations, leading to a shortening of the Debye length and thus a decreasing importance of any stabilising electrostatic interactions [118]. Methods for reducing adhesion forces in microrobotics are based around reducing contact area between objects [112,119]. The reason for this can be easily appreciated when one considers the equations for the van der Waals force between a sphere and a plane (11) [120]—a key contributor to adhesion forces.
(11)$$F_{vdW}=\frac{H}{6}\frac{d}{2z}+\frac{d}{2{\left(z+d\right)}^{2}}-\frac{1}{z}+ln(\frac{1}{z+d}).$$

In Equation (Equation 11), *H*represents the Hamaker constant of the material [121], *d*is the diameter of the microsphere and *z*is the separation between the sphere and the plane. This equation assumes the same material for the plane and the microsphere, and that the plane is extremely large compared to the microsphere, and so has some limitations, but it clearly illustrates the effects of separation distance and size of the objects on the van der Waals forces between them. Therefore, it can be deduced that adhesion forces can be limited by increasing separation between moving parts of the nanomachines, and by minimising the contact area between parts, which is supported by results from the literature.

Another challenge in creating multi-component optical microrobotics is being able to use non-spherical objects. Relatively small (d ≈λ) spherical shapes are the easiest objects to trap with single Gaussian traps, as they are isotropic and the curved surface leads to lower reflection and so to lower scattering forces when compared to flatter objects. Additionally, as already mentioned, they are also the easiest shape to model. Conversely, objects with high aspect ratios, such as cylinders, tend to orient themselves with the axis of the optical trap [122], a tendency first discovered when Ashkin and Dziedzic trapped cylindrical viruses [17], which can lead to difficulty in manipulating such objects as desired. A method for getting around this problem is to add spherical features as trapping “handles” [123], similar to the use of functionalised beads in DNA stretching. This strategy has been adopted by many researchers, and spherical parts feature in many optical microrobot designs [23,25,112,124]. Another strategy is to use several optical traps, positioned so that the forces are balanced when manipulation takes place, regardless of the shape of the “handles”. Examples of this can be found in the use of cylindrical trapping points in the development of an optical scanning probe [125], where spheres are used as tracking points rather than trapping handles, and in the use of flat discs as handles for structures produced using 2D deposition and etching methods [99].

While the gradient force is most commonly used to control the movement of optical microrobots, optical tweezers can also actuate micromachines through the scattering force, or through thermal effects. The use of the scattering force to trap objects has perhaps most successfully been used in the case of the counter-propagating optical trap, where two aligned, counter-propagating laser beams work together to create a balanced trap based on the scattering force produced by each laser [126,127]. This configuration, while complex compared to single-beam optical tweezers, allows for lower intensity light to be used, and so does not require the high NA objectives used for a gradient trap. The use of a lower NA objective also allows for longer working distances than optical tweezers, which are generally limited to a working distance of less than a millimetre from the objective. On the other hand, an example of the use of thermal effects from optical tweezers to actuate microtools is the creation of a “micro-syringe”, where the inclusion of a thin metal layer inside a 3D printed chamber creates thermally-induced flow in response to the laser light, moving silica and polystyrene microspheres in the process [128]. The process of pulling in a microsphere to later be ejected can be seen in Figure 3.

As discussed previously, the force resulting from the interaction of the laser beam with the object of interest is commonly split into two parts, and $F_{grad}$ > $F_{scat}$ is commonly accepted to be the condition necessary for stable trapping. However, the scattering force can also be used to actuate microrobots, and can be used for applications involving a rotating part which can be “pushed” by the light, rather than being dragged into place [129,130]. In-plane rotation of a twin-rotor micropump has been demonstrated, by focusing a Gaussian beam on the sides of flat-lobed “wings” [131]. Out of plane rotation has also been demonstrated using an optical “paddle wheel”. In this case two micro-spherical features were used as “trapping handles” to hold the object in place, while the central “paddle” was driven by the scattering force of another trap, creating out-of-plane rotation [132]. A schematic of this microrobot, as well as stills taken from a video of it moving, can be seen in Figure 4.

Optical forces have also been utilised to rotate chiral objects, where the design of the object means that the incoming momentum is unbalanced and creates an optical torque around the beam’s trapping axis [133]. This has been utilised by researchers to produce micropumps, through the rotation of self-orienting Archimedes screws [134], and through the use of rotors with tilted “blades”, similar to a waterwheel [135]. Anisotropic quartz particles have been used to apply and measure torque from Gaussian, single-beam optical tweezers, due to the birefringence of quartz resulting in non-uniform response to linearly polarised light [136]. Another paper by the same group demonstrated the influence of changing the beam polarisation on the particle’s resulting torque, showing that the torque obtained depends on the polarisation rotation rate, allowing for the use of a trapped particle as a passive constant-torque wrench [137]. In the case of optical rotors, it is generally true that the rate of rotation is governed by the power of the laser used, and these could be used for micropump applications such as cell-sorting or as mechanisms for exerting mechanical force on objects such as cells, to investigate mechano-transduction processes.

Control of articulated optical microrobotics is commonly achieved by manually moving them using an optical trap. Therefore, the development of automated control strategies presents a gap in the research, although some examples exist for simple microrobotics set-ups involving rigid bodies [138,139]. Improving the control strategies available for articulated optical micromachines, particularly ones that allow for parallelising and automating, would help to move them from research curiosities to useful tools for non-experts. The ability to do this is strongly linked to the ability to repeatably produce micromachines, as tiny differences in structure can have large effects at the microscale, which could impact functionality. This was noted in Reference [112], where success rates of different designs were tracked in order to determine a reliable microrobot design. The ability to model the behaviour of microrobots in response to the beam is also important for control system design, and repeatable experimental results are integral to building reliable models. However, a major difficulty that researchers face is the fact that all parts of an optical microrobot are affected by the presence of an optical trap. This means that there is difficulty controlling the movement of multi-component microrobots, as the different components are all attracted to a single point, the optical trap. Therefore the shape of the micromachine’s components, and the mechanical coupling of the different parts is key to ensuring the micromachine operates as required. Additionally, as covered earlier in this review, the number of optical traps that can be used, and the ability to control them separately should be considered.

## 7. Effects of Microrobot Shape

As previously touched on in this review, the simplest shape for optical trapping is the homogeneous sphere. However, obviously this is not an ideal shape for multi-body microrobots, for tasks involving gripping, or for tasks where movement around a fixed axis is the desired outcome. Therefore, the shape of the microrobot must be decided according to the same principles as any other engineering challenge: namely, the task at hand and equipment restrictions with respect to manufacturing and controlling the resulting machine.

The production of simple, classical machines involving lever arms, springs and even Archimedes screws has been achieved using TPAP, and functionality has been demonstrated by using optical tweezers to actuate movement. The shape of these microrobots is highly relevant when it comes to their functionality. In the case of an optically controlled lever arm [24], the long arm, if unconstrained by the pin-joint, would naturally orient itself with the trapping beam. The fixed axis, which is attached to the substrate, prevents this from occurring to some extent, but the effect has been noted as a restriction of micro-scale pin joints, as it increases undesirable, out-of-plane movement [25]. The inclusion of spherical features which act as “trapping handles” serves to reduce undesirable motion, as these points are more strongly drawn into the optical trap than relatively flat surfaces.

While this review has already covered some of the situations in which the scattering force is useful for actuating a microrobot, the shape of micromachines that are intended to be actuated in this way also deserves some attention. For instance, the angle of interaction of the optical tweezers with the surface of the screw is vitally important for producing rotation in the desired direction. This has been demonstrated in the creation of the wheel with tilted blades referenced earlier [135], where the direction of rotation was changed by moving the optical tweezers to different points of the wheel. This effect was exploited in the creation of a twin-rotor using a similar design by different researchers [140]. This can be simply explained according to the refraction and reflection of light in an asymmetric shape leading to the resultant forces being directed off-centre and leading to an optical torque [133]. This concept also explains the positive impact of adding spherical “trapping handles” to micromachines, as the curved surface means that the refractive (gradient) and reflective (scattering) components of force from the optical tweezers are directed through the centre of the sphere.

While spherical handles work well for optical tweezers utilising Gaussian beams, where the refractive index of the micromachine is higher than that of the surrounding medium, this is not the case for situations where the refractive index of the object is the lower value of the two. In this case, in fact, the opposite becomes true, and doughnut-shaped objects have been stably trapped using Gaussian optical tweezers, utilising the interaction of the gradient force on the inner walls of the doughnut [141,142]. A diagram of this can be seen in Figure 5, where the case of a spherical particle (a) is compared with ring-shaped particles (b) and (c), and the impact of the beam waist position in the Z-axis is evaluated. The demonstration of stable optical trapping of low-refractive index particles further increases the options for optical trapping studies by increasing the range of mediums that can be used, and it is possible that ring-shaped objects could be used as “trap handles” for micromachines in these scenarios.

## 8. Choice of Optical Tweezers for Microrobotics

It can be seen that the ability to create multiple optical traps is extremely useful in microrobotics. This is particularly true in the case of free-floating micromachines with moving parts [112,132], when the overall position of the microrobot must be controlled as well as its relative motion. Using several different laser beams would be both expensive and greatly increase the space required for an optical trapping set-up. Luckily, several solutions have been developed for the creation of multiple traps from a single focused laser beam. These can be divided into space-sharing and time-sharing set-ups, with holographic optical tweezers [143] being an example of the first type and fast-switching acousto-optic deflectors [144] being an example of the latter. Holographic optical tweezers (HOT) utilise spatial light modulators (SLM) to split an incoming beam into several different beams, which are then directed through the microscope aperture, as shown in Figure 6. This technique is particularly flexible in terms of the number of traps that can be created, and the range of movement that is allowed, as it relies only on the pattern imposed on the SLM. Work regarding the calibration of HOT using the power spectrum analysis has also revealed that trap stiffness does not vary significantly during trap movement, and that trap placement is controllable with single-nanometer resolution [145]. These capabilities identify HOT as a useful and versatile tool for force application and measurement in biophysical experiments. However, the real-time use of the technique is limited, due to the calculations required to determine the required pattern of orientations of liquid crystalline pixels across the device in order to create the desired trap positions. Several different methods for performing this calculation are presented in the literature, including the high-performance yet low-speed Gerchberg-Saxton algorithm [146] and the low-precision yet high-speed direct superposition method [147]. Therefore, when calculating holograms for optical tweezers or other laser-based applications, a trade-off must be made in terms of either speed or accuracy [148], with attempts made to adapt these existing algorithms to produce high speed, high performance alternatives [149,150].

Time-sharing set-ups for optical tweezers are based on rapid movement of the beam and typically use acousto-optic (AOD) or electo-optic deflectors (EOD), which deflect the beam by a certain angle, controlled by the frequency of an input signal [151,152]. The calculations involved in this type of dynamic optical tweezers are less time-intensive than those for HOT, but time-sharing optical tweezers are limited in the displacement of traps that can be achieved. This limitation has been tackled through the introduction of additional deflectors [153], and through careful selection of the transducer used to convert the signal that controls the deflector position, which can affect both resolution and range of frequencies that can be used [154]. As HOT and time-sharing dynamic optical tweezers both have their shortcomings, and both have been used successfully in optical microrobotics applications, the decision to use one or the other is largely based on hardware restrictions, with the introduction of an SLM into an existing general optical tweezers set-up potentially much simpler than the creation of a beam-deflecting system. Additionally, commercial options are available for both HOT and AOD-based optical tweezers, but in laboratories where a small range of specific experiments are performed then it may be preferable to build a custom optical trapping set-up according to specific requirements, and researcher expertise. Additionally, many advances have been made in the development of novel optical tweezers setups, and while they are not a central part of this review it is worth mentioning the counter-propagating optical trap [99,127], and the optical fibre tweezers [155]. The latter allows for light to be precisely directed into a sample by use of optical fibres which are pig-tailed to the laser diode, focusing the light using the tapered end of the fibres. Optical fibre tweezers offer the opportunity to precisely trap objects in crowded samples, and so could be considered for situations where trapping is to take place in a complex environment. A review of optical fibre tweezers that covers the use of the technology for a range of optical trapping applications is provided by Zhao and colleagues, including a theoretical explanation of both dual and single fibre setups [156].

## 9. Looking to the Future

There is potential for the use of micromachines in almost any discipline where optical tweezers are used to perform tasks, rather than as a laser-physics demonstration. Also, as previously mentioned, the improved and simplified ability to produce optical microrobots without post-manufacture assembly means that researchers from a variety of backgrounds can make use of the opportunities that optical microrobots offer. However, the achievements in microrobotics to date appear to have a bent towards biological research, where the dexterity and precision of optical microrobotics could make activities such as polar-body biopsy or cell enucleation swifter and easier. One of the main reasons that microrobotics have such potential in these areas is the potential for parallelising experiments. For instance, cell aspiration is typically performed using a very thin micropipette, which makes it a delicate task that is susceptible to any disruption to the apparatus [157]. Using a micro-machine that features a sharp point, similar to the nanowire used for temperature measurement in Reference [24] to probe cell stiffness could improve experiment success rates by reducing equipment breakage and decoupling the aspiration set-up from outside vibration, and this concept is shown very simply in Figure 7. Using multiple micromachines controlled by multiple optical traps could increase the speed of such experiments, with many aspirations taking place in parallel with one another. Additionally, the highly customisable and controllable nature of optically controlled microrobots in terms of output force and object displacement means that experiments adapted for optical tweezers using optically controlled microrobots could have lower intrinsic uncertainty than conventional, non-optically actuated methods. A potential example could be single-cell surgery, where the exact force required to breach the cell membrane without causing undesirable damage could be carefully applied with optical microrobots every time. As well as this, the potential that optical microrobots have for reducing laser-induced damage during optical tweezer assays further cements their probable application in biological research in the future, as these assays get adapted to the technology. An example of this could be micro-machine assisted DNA stretching, by adapting the protocol that already makes use of functionalised microbeads, as discussed earlier in this review.

The methods developed to control optical microrobotic systems could also indirectly benefit research using optical tweezers. For example, the work done to determine position and orientation in a 3D workspace [158], or to control the position of holographic optical traps in real-time [139,159] has applications in optical tweezer-based cell sorting. Therefore, it is fair to say that research into optical microrobotics will offer both direct and indirect benefits to the scientific community, moving forward.

## 10. Conclusions

The use of optical tweezers in biological studies has allowed for the investigation of processes and subjects which would otherwise have remained out of reach, due to the resolution of force and displacement possible. However, as has been outlined in this review, the negative effects of optical tweezers are well-known. The common methods for reducing optical damage, such as altering trapping wavelength or introducing radical scavenging species do not allow for flexibility in the experiments that can be performed, and may require completely changing the trapping set-up. This means that such methods are unsuitable for retrofitting to existing trapping set-ups, and are not suitable for general trapping set-ups which may be used for a variety of applications where wavelengths that are more damaging to biological samples are completely acceptable. Reducing the power of the laser is also often less than ideal, as this lessens the forces that can be applied, which limits the ability to research high-force phenomena such as DNA overstretches. Optical micromachines present a flexible solution to these problems, as they can be used to limit direct exposure of the subject to the laser, which has been theorised to be one of the main ways to reduce optically induced damage, through direct sample heating, photobleaching and generation of free radicals in close vicinity to the sample. While indirect manipulation through microrobotics will not remove the risk of damage due to free-radical damage, it would move the origin further away from sensitive biological matter, such as DNA, resulting in more time before their diffusion to the polymer. Additionally, optical micromachines show potential to be used to extend the capabilities in biological studies, with proof of concept demonstrated for force amplification and direct force measurement. The resolution that can be achieved through two-photon absorption polymerisation, along with the ability to print articulated machines also means that many different micromachine designs are possible. This not only means that the technology is extremely flexible and can be adapted for many different applications, but it also means that there could one day be solutions for many of the challenges associated with the technology, such as the dominance of adhesion forces. However, there is still much work to be done in making micromachines attractive to non-roboticists, such as in automating manipulation and improving repeatability. Overall, optical micromachines present many opportunities for biological studies, as well as interesting engineering challenges, and will likely be a subject of research interest for years to come.

## Figures and Tables

**Figure 1 micromachines-11-00192-f001:**
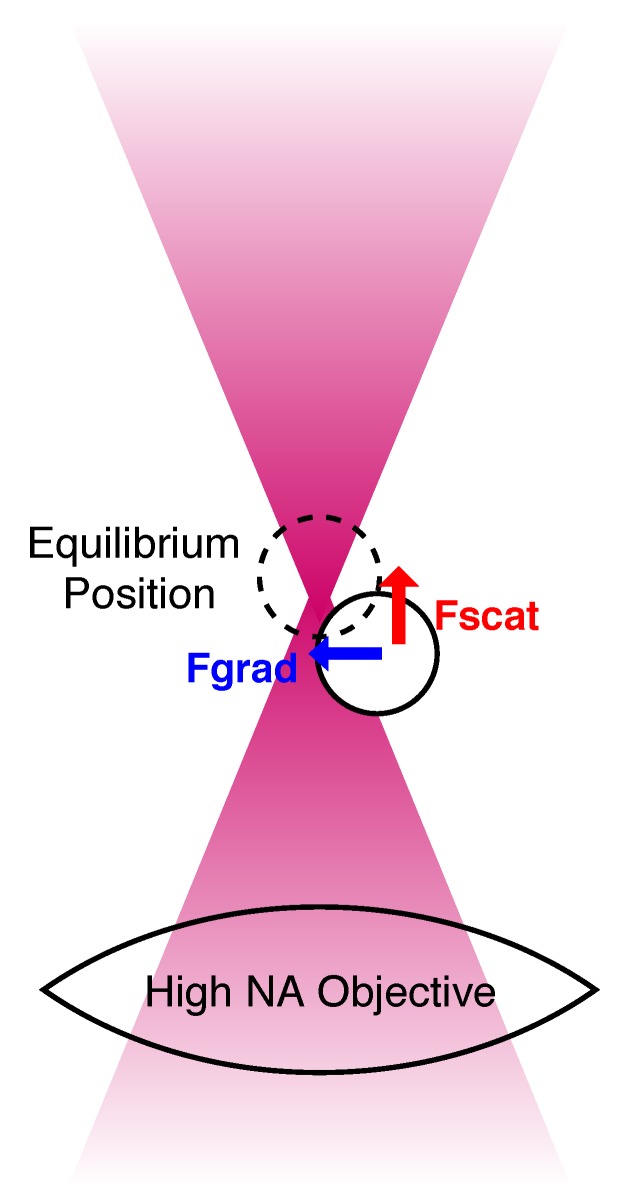
Optical tweezer theory treats the force associated with the focussed laser beam as two components, which can be visualised as resulting from the intense focussing of the light.

**Figure 2 micromachines-11-00192-f002:**
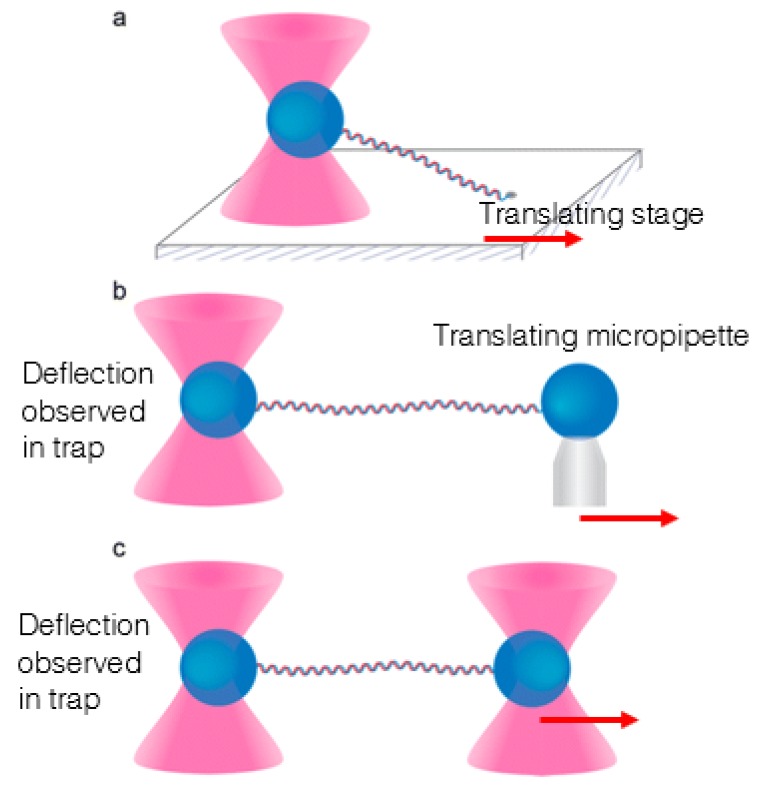
Three different methods for stretching a DNA tether with optical tweezers can be seen in this figure: the translating slide method (**a**), using an anchoring micropipette (**b**) and using two separate optical traps (**c**). Reproduced with permission from Reference [75]. ©2014 American Chemical Society.

**Figure 3 micromachines-11-00192-f003:**
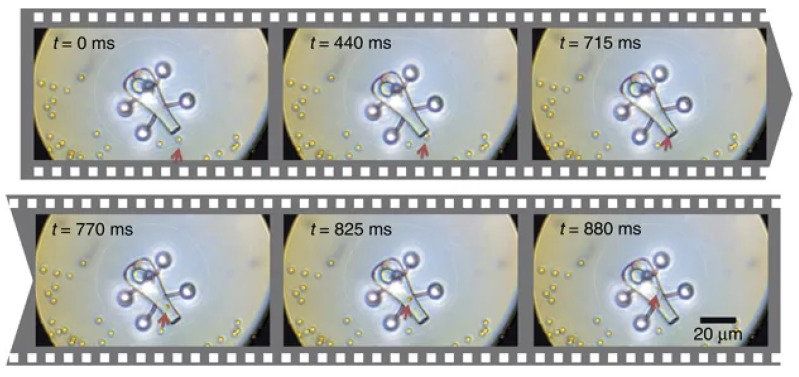
The inclusion of a thin metallic layer inside a 3D printed chamber leads to the movement of silica and polystyrene payload particles through convection. Reproduced from Villangca et al. [128], under Creative Commons license NC ND 4.0.

**Figure 4 micromachines-11-00192-f004:**
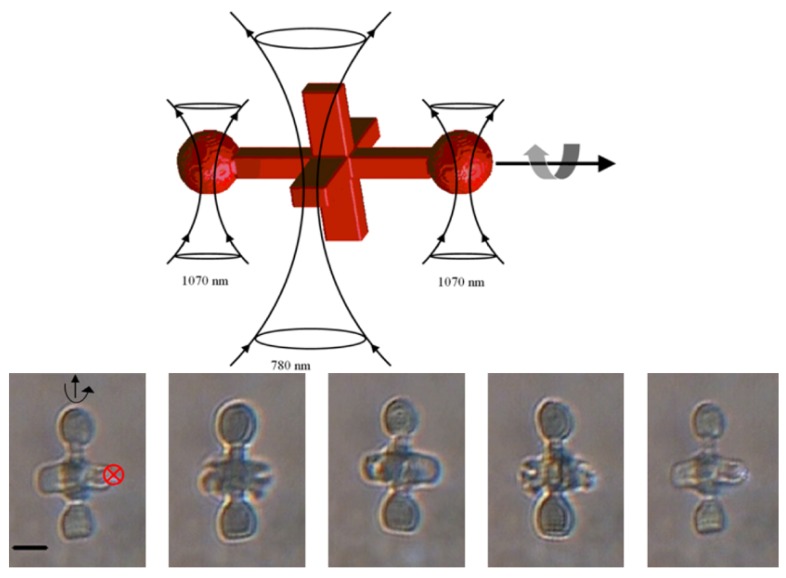
Asavei et al. produced an optical “paddle wheel” held in place using two 1070 nm optical traps and driven with a separate 780 nm beam. Reproduced from “Optically trapped and driven paddle-wheel”, [134], under Creative Commons license CC BY 3.0. (**Top**): Schematic of the device. (**Bottom**): Stills taken from a video of the device in action, where the position of the 780 nm beam is shown with a red x.

**Figure 5 micromachines-11-00192-f005:**
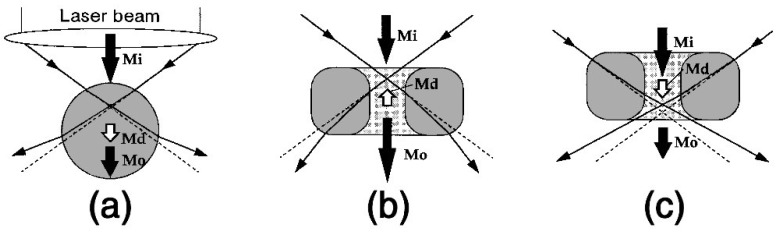
Illustration of the interaction of the optical tweezers with differently shaped low-refractive index particles, and the impact of the position of the beam waist. In (**a**) and (**c**) the object shape and the beam waist position act to push the object out of the trap, whereas in (**b**) the incident forces are balanced on the inner walls of the ring, and the position of the beam waist in the Z direction acts to pull the object towards the waist, producing a stable trap. *Mi* is the incident momentum, *Mo* is the outgoing momentum and *Md* is the momentum transferred from the light to the object. Reproduced with permission from Reference [141] ©The Optical Society.

**Figure 6 micromachines-11-00192-f006:**
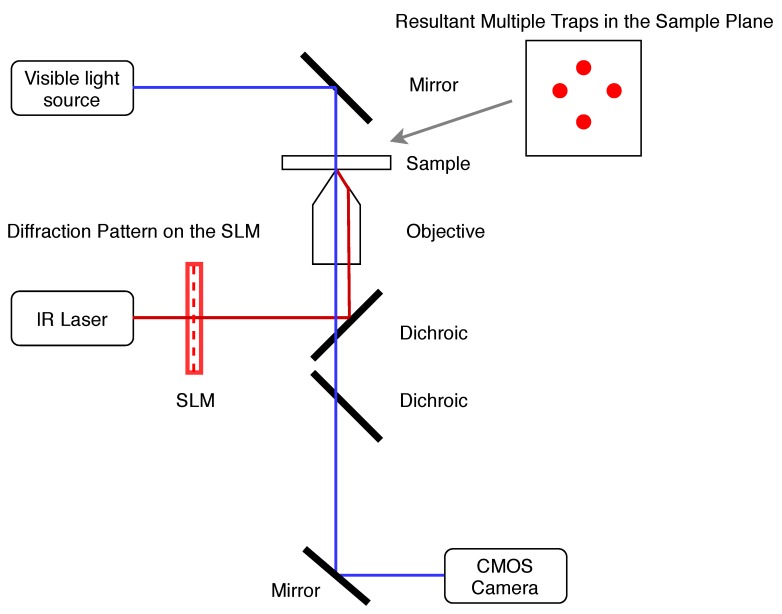
Simple schematic of holographic optical tweezers, based on the authors’ laboratory set-up.

**Figure 7 micromachines-11-00192-f007:**
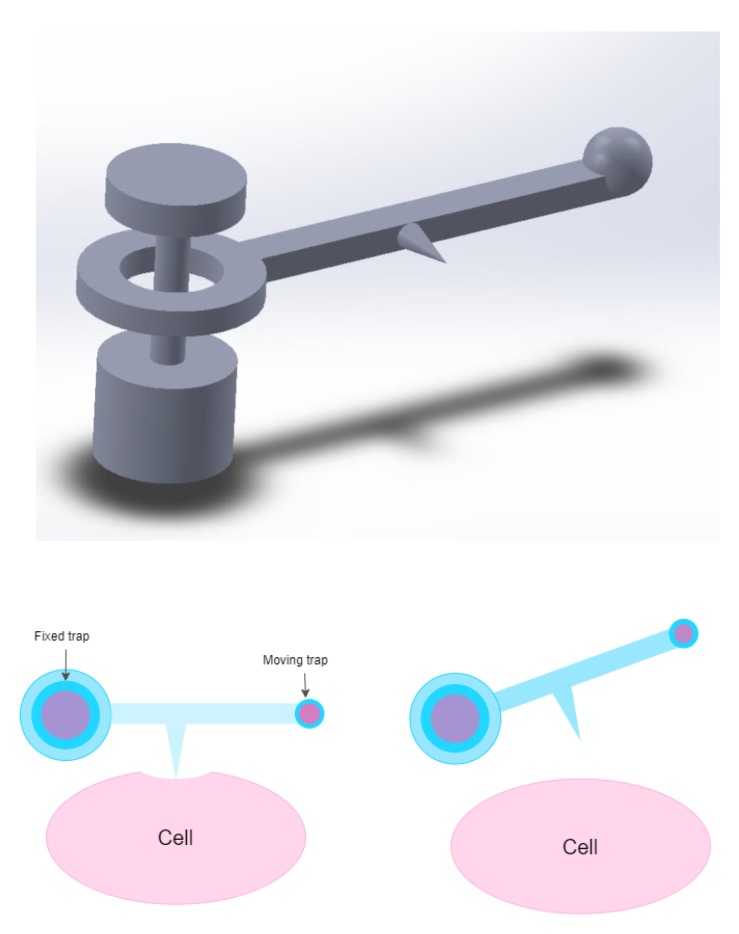
A free-floating microrobot, such as this lever, equipped with a sharp tip, could be used to apply amplified forces for measuring cell membrane stiffness. At least two traps are necessary for this, one fixed trap to hold the microlever in place near the cell and another to move the lever arm.

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
