# Peer review of "Optical Micromachines for Biological Studies"

_micromachines, 2020, doi:10.3390/mi11020192_

Round 1
Reviewer 1 Report
The manuscript by Phillippa-Kate Andrew reviews the optical tweezers in biological research, especially for the concerns in optical trapping for the samples and optical microrobots. The authors started with the basic theory of optical trapping which is easy to follow for the readers. Then they detailed the concerns of optical tweezers in biological studies. They also give a very good overview of the microrobots with plenty of references.
The manuscript is clearly written and comprehensive and should be of interest to the readers of Micromachines as a review paper. The review as presented has limitations, especially in scope, but these can be referred to other related review articles.
Author Response
Thank you for your encouraging comments and for taking the time to review the paper.
Reviewer 2 Report
The authors did a great work to review the application of optical tweezer in biological study. It's a great pleasure to read this article. The writing style and the way the manuscript is organized is logical. The authors have covered the history and a majority of shining moments in the history of optical tweezers. Therefore, I highly recommend accepting this paper for publication. However, I have a few suggestions to the authors that may help to further improve the paper:
I understand it is challenging to includes everything in one paper, but it is still beneficial to include some other tappings such as a counter-propagating trap (Optics Express, 25, 2496 (2017) and traping with optical fibers. The counter-propagating trap allows to reduce of the trapping power and trap objects that cannot be trapped with a single-beam trap. The trapping with optical fibers along to simplify the trapping system.
I am not saying the authors should spend many pages on these suggested topics. but I think including these methods will a plus for this paper. The authors can briefly introduce these methods.
Author Response
Thank you for the feedback and for the suggestion. Please see detailed response in the attached pdf.

Reviewer 3 Report
This is a comprehensive review article, built around the theme of using microscale devices to manipulate biological samples, where the microscale devices themselves are manipulated in optical tweezers. The article is quite wide-ranging, such that it almost encompasses three sub-reviews: on the theory of optical trapping; on the potential for optical trapping experiments to result in damage of biological samples; and experiments using micromachined devices in optical tweezers. Each of these topics includes copious references for the interested reader, meaning that the article should serve as a useful go-to reference for those entering one of these sub-fields.
Given the wide-ranging nature of the article, I would have found it helpful in the introduction to have more of a roadmap to the content; the introduction is quite short. Perhaps the authors could include some illustrative schematics for how they envision micromachines being implemented for biological sample manipulation. Such an illustration is lacking in the article, and hence, it feels more like a collection of topics around a central theme, rather than a truly visionary review (which I think this has the potential to be, with a bit of imagination included and conveyed through illustrations of potential applications). This is my only significant suggestion for improvement: please give me a better sense of what you think these optical micromachines might be used to do!
Other minor suggestions for improvement follow.
In a few cases (e.g. 223, 260, 537) I found the authors’ terminology around “handles” for DNA experiments to have the potential for confusion, since the term “DNA handles” is itself conventionally used to describe double-stranded DNA added to the ends of proteins or RNA to link to beads for unfolding experiments. Coming from the biophysics and soft matter community, I would not have thought of directly trapping DNA in any case. If the authors add some figurative illustrations, as suggested above, then I think this point about beads and DNA would be much clearer and avoid any potential confusion.
Line 84: In the Rayleigh regime, don’t particles become polarized (and hence, act as dipoles), rather than charged, in response to the EM field?
Line 136: The authors could include other methods of force detection (e.g. momentum determination, Farré and Montes-Usategui, Opt. Express 2010).
Line 160: I was very surprised that the authors did not cite the original WLC model papers, and would strongly suggest that they do so. (Bustamante et al., Science 1994 and Marko & Siggia 1995)
Line 169: Clarify that “unwanted coupling” is mechanical coupling (if this is indeed the case)
Line 318: Two-photon absorption scales nonlinearly with intensity because of the requirement for two photons (rather than one), not because of the speed of absorption (as implied by this sentence).
Line 330: I found the discussion around the type of resin to be somewhat confusing. Here and elsewhere, it is not made clear whether these “micromachines” are to be made in-situ (using the trapping laser beam for etching/curing) or elsewhere. I had been under the impression that the micromachines discussed in this review were all made elsewhere and then transferred to the trapping chamber, but realized that this was not explicitly discussed. The discussion here around absorption of the resin by the trapping wavelength made me question this assumption.
Line 393: References to the trapping of micro-cylinders by the group of Michelle Wang (first quartz, more recently silicon nitride) could be added here. The optical torque wrench of Michelle Wang’s group (quartz microcylinders) could also be added to the discussion and references in the paragraph starting line 417.
Figure 6: Without an illustration of the in-plane results of holographical optical tweezers (i.e., showing multiple traps generated from a single beam), this figure does not communicate any useful information.
Lines 501-515: Given the authors’ discussion in other parts of the paper regarding force calibrations, it would be useful to include the work of Nancy Forde and colleagues on force calibration of holographic optical tweezers.
Line 533: It is not clear to me how optically controlled microrobots could have “lower intrinsic uncertainty than conventional methods”. Obtaining sub-nanometer spatial resolution with spherical beads in conventional optical traps is pretty impressive!
Line 557: Only some problems with laser exposure could be solved with micromachines, as for example radical generation will presumably still occur. A reference to the “theorised to” statement could also be provided.
Author Response
Thank you for taking the time to review the paper, and for providing detailed and constructive feedback. Please see the attachment for detailed response.
